# Enterprise Alexandria: Online High-Precision Enterprise Knowledge Base Construction with Typed Entities

**John Winn, Matteo Venanzi, Tom Minka, Ivan Korostelev, John Guiver**[*]**, Elena Pochernina, Pavel Myshkov, Alex Spengler, Denise Wilkins, Siân Lindley, Richard Banks, Sam Webster, Yordan Zaykov**

*Microsoft Research, 21 Station Road, Cambridge CB1 2FB, UK*

{JWINN, MAVENA, MINKA, IVKOROST, ELPOCHER, PMYSHKOV, ALSPEN, DWILKINS, SIANL, RBANKS, SWEB, YORDANZ}@MICROSOFT.COM, [*]JOHN.GUIVER@OUTLOOK.COM

## Abstract

We present Enterprise Alexandria, one of the core AI technologies behind Microsoft Viva Topics[1]. Enterprise Alexandria is a new system for automatically constructing a knowledge base with high-precision and typed entities from private enterprise data such as emails, documents and intranet pages. Built as an extension of Alexandria [Winn et al., 2019], the key novelty of Enterprise Alexandria is the ability in processing both the textual information and the structured metadata available in each document in an online learning fashion, making use of any manual curations that have happened in the interim. This task is performed entirely eyes-off to respect the privacy of the user and the restricted access to their documents. The knowledge discovery process uses a probabilistic program defining the process of generating the data item from a set of unknown typed entities. Using probabilistic inference, Enterprise Alexandria can jointly discover a large set of entities with custom types specific to the organization. Experiments on three real-world datasets show that the system outperforms alternative methods with the ability to work effectively at large scale.

## 1. Introduction

As the volume of textual information produced within organizations continues to grow dramatically [Radicati and Levenstein, 2019], it is critical for Automated Knowledge Base Construction (AKBC) systems applied to enterprise scenarios to process data *automatically* and at *large scale*. This is becoming a key research area in enterprise knowledge discovery [Loshin, 2001] where a key task is to turn unstructured text present in emails, documents and intranet pages into a single, continuously available, consistent and up-to-date knowledge base. In doing so, due to the restricted access of enterprise documents that are continuously edited by the users, the entire knowledge base must be constructed *incrementally* and *eyes-off* (i.e., documents are not accessible for labeling) for each organization [Voigt and Von dem Bussche, 2017]. Furthermore, since any automated system will inevitably have gaps in the extracted knowledge, the management of such systems also involves the role of *knowledge curators* acting as a trusted source that can editorially curate or expand the mined knowledge [Maedche et al., 2003]. Therefore it is important for an AKBC system to be designed to accommodate and learn from curated content. The aim is to achieve a

---

1. www.microsoft.com/en-us/microsoft-viva/topics

virtuous cycle where people and machines continually work together to keep the knowledge accurate and complete.

A key characteristic of enterprise documents is their structured nature, including both structured text and associated *metadata*. For instance, emails have a subject, sender, list of recipients and attached documents while documents have a creator and a list of editors, viewers and commentators as well as structure within the document, such as sections and titles. We argue that both kinds of information are critical to build an accurate knowledge base. In this respect, there is a wealth of knowledge discovery methods that process text without explicitly modeling metadata [Yamada et al., 2020, Devlin et al., 2019, Zhang et al., 2019]. Other systems can leverage the full data structure if some amount of training labels are available to infer the relationship between each structured field (e.g. email sender) and the entity properties (e.g. project lead) [Zhang, 2015]. Unfortunately, the privacy restrictions of enterprise data do not allow for their labeling by human annotators, making these methods unfeasible.

We address this gap by developing an automated system, Enterprise Alexandria (EA), as an extension of Alexandria [Winn et al., 2019], an existing system designed for Web-based information extraction. EA is able to jointly discover entities and their associated types with very minimal supervision in compliance with the privacy requirements of enterprise data, which makes it the first entity discovery method applicable to enterprise scenarios to the best of our knowledge. In EA, the probabilistic program is extended to generate both the document and its metadata from a set of latent entities. Inference is partly performed automatically using an efficient message-passing algorithm [Minka et al., 2018], and partly with a handwritten scalable algorithm that can process a large set of documents in batches and build the knowledge base incrementally, making use of any manual curations that have happened in the interim. Furthermore, EA adds an important ability to learn the set of types associated with each entity (e.g. product, event, team, project) from a pre-defined type hierarchy. In order to ensure that the type hierarchy is sufficient, EA is also able to semi-automatically discover type names from enterprise data. The importance of determining the types of extracted entities receives support from a study of workers in two organizations, which showed that a wide variety of knowledge types are important for work, and that workers seek to engage with these different types in different ways. We argue that types are needed not just generically, but to support work in particular industries and organizations. Using real-world datasets, we show that the quality of the entities and their associated types extracted by EA outperforms the baselines by significantly increasing the precision of the extracted knowledge.

In summary, this paper provides the following contributions to the state of the art: (*i*) Findings from a qualitative study of members of two multinational organizations, which highlights how entity types are important for work. (*ii*) An extended probabilistic program that improves the existing Alexandria model and can perform entity discovery jointly from structured text and the associated metadata. (*iii*) A new mechanism for updating the discovered knowledge incrementally with the ability to support curated content contributed by human users. (*iv*) A real-world evaluation showing that EA performs efficiently at scale while significantly improving the precision of the extracted knowledge compared to four baselines on several real-world datasets.

## 2. User Study on Knowledge Categories

To best inform the design of our model, and to gather data on real entity types as a baseline for our evaluation, we undertook a qualitative study with prospective end users who were employed in two different industries. In the study we asked: How do organization members think about knowledge types, and what types of entities need to be discovered to support everyday work? The study was conducted in two workshops with 12 (5 = men, 7 = women) knowledge workers who were employed by one of two multinational organizations (6 participants in each). The first organization was a large professional services firm; the second was part of the travel and tourism industry. Participants came from a variety of backgrounds in terms of role (e.g. designer, program manager, analyst, information architect) and ranged in seniority from junior to national leadership. In order to provide context, participants were first shown a short video that described how an automatically-generated knowledge base might support work. Participants were then asked to think about the different components that make up their work, which they might want represented in the AKBC system. Participants were given four examples of such components (project, budget, product launch, designer) in order to stimulate ideas. The study received ethical approval from the host Institutional Review Board (IRB). Each participant provided informed consent for data collection and use prior to participation.

| Superordinate class *(researcher generated)* | Knowledge Types *(user generated)* |
|---|---|
| Seed types (examples given to the participants) | Projects, Budgets, Designers |
| Technology | Data, Analytics, Algorithms, Technology, Architecture, Service, Service measurement, |
| People | Headcount, Teams, Stakeholder, Users, Contacts, Communicate, Social Support Community, Beta Testers, Build network |
| Digital Artifacts | Financial information, Roadmap, Vision, Strategic plan, UX Research, Design, Documentation |
| Internal media | Brand, Templates, Learning materials |
| External media | Media, Research papers/reports |
| Internal containers | Design library, Research templates, Support model handbook, Device catalog |
| Insight | Problems, Best Practices, Industry Knowledge, Idea lifecycle, Demand, Management Buy-In, Brain |
| Tools | Tasks and To-do, Checklist, Tooling, Microsoft Office 365, Benchmarking, Enhancement |

Table 1: The knowledge categories provided by the participants of the user study.

Table 1 shows the 48 unique knowledge types generated by participants in total, which we manually grouped bottom-up (via an affinity diagram) into eight superordinate classes to help summarize results. The knowledge categories were grouped based on the explicit meanings of the category names, descriptions and discussions provided by participants in the workshop. While three of the knowledge categories reported by participants were confirmations of the examples we gave to participants (project, budget, designers), the remaining 45 categories were generated by participants independently. Rather than suggesting that all of these categories are necessary for every knowledge base, we suggest these categories capture this set of prospective users' *aspirations* for a knowledge base. Overall, this qualitative study demonstrates why types matter: a wide variety of entity types are important for work, and workers seek to engage with these different types in different ways. In the following

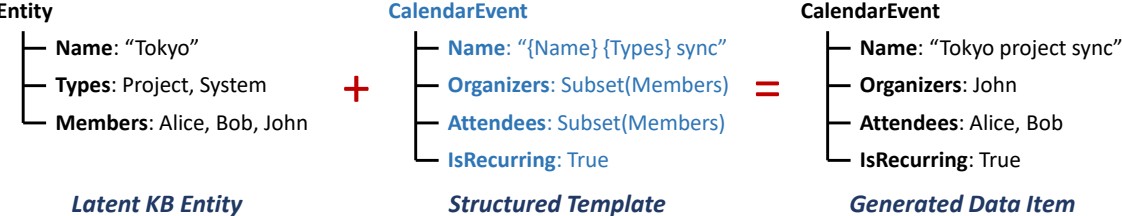

Figure 1: An example of a structured template being applied to a knowledge base entity (a project named "Tokyo") to generate a data item, in this case a calendar event.

discussion, we move on to describe a model to learn the entity types automatically as part of the AKBC task.

## 3. The Enterprise Alexandria Model

The Enterprise Alexandria model is an extended form of the probabilistic program used by Alexandria [Winn et al., 2019]. The main extensions are described below.

### 3.1 Structured Templates for joint processing of Text and Metadata

The Alexandria probabilistic program uses a set of templates to generate text. EA extends this program to generate entire data items, such as emails or calendar events, using manually-specified *structured templates*. Figure 1 illustrates how a structured template can be used to generate an observed data item from a latent knowledge base entity. In this example, the structured template is applied to a Project entity named "Tokyo" to create a recurring calendar event 'Tokyo project sync' with suitable organizers and attendees. During inference this process is inverted, so a compatible calendar event will be matched against this structured template to produce the knowledge base entity. A structured template consists of the type or types of data items which can be generated (such as emails, calendar events or documents) along with a set of properties for the generated data items each with a suitable value generator. Value generators can be one of three kinds: (*i*) **A template-based generator** which generates a string given a template. If more than one template is provided, one is selected at random. The Name property in Figure 1 uses this kind of value generator. (When 'Types' is used in a template, one type is selected at random.) (*ii*) **A subset generator** which generates a collection value as a random subset of a specified collection in the entity. The Organizers and Attendees property in Figure 1 use this kind of generator to generate subsets of the Members property. (*iii*) **A constant value generator** which always gives a particular value. The IsRecurring property in Figure 1 uses a constant value generator which always returns 'True'.

### 3.2 Entity Names and Variants

In the original Alexandria model, the prior probability of an entity name was uniform over the set of valid names. EA uses a more sophisticated model (tuned on Wikipedia titles)

which takes into account the number of words, the length of each word, and more. This means that two mentions of the same long name are less likely to be a random collision and more likely to be referring to the same entity than two mentions of the same short name. This improves the ability of the system to disambiguate mentions of different entities sharing the same name.

In enterprise data, it is common for an entity to be referred to by different variations of a name, such as abbreviated forms, capitalizations, etc. EA allows an entity to have alternative names, as long as the alternatives are compatible variants of each other under a set of variants models. The supported variants models are:

- **Case and diacritic** variants where names differ only in case or in the presence/absence of accents or diacritics;

- **Separator** variants where the names differ only in separators (such as '&' or '+') or the separators are removed entirely;

- **Name phrase** variants where a name "Tokyo" and a name phrase "Project Tokyo" can be identified;

- **Acronym** variants where one name is an expansion and the other an acronym or partial acronym.

When computing the probability that two mentions refer to the same entity, if their names do not match but are compatible alternatives, then the probability is the same as if the names matched (on the more probable name), times a constant penalty.

### 3.3 Type Discovery

As in Alexandria, facts about an entity are represented by named, typed properties. EA includes a property named 'Types' which holds the types for the entity, selected from a predefined set. EA semi-automatically learns this set of allowed types through a process of *type discovery*. This process starts with a small manually-provided seed set of types, such as {Project, Team}. New types are then added iteratively as follows: (*i*) Entity discovery is run to extract entities from a set of enterprise data items, given these seed types. (*ii*) Fact retrieval is run using the names of these discovered entities, as described in Winn et al. [2019]. However, the schema is modified to allow values of the Types property to be any string of 1–3 words, rather than one of the fixed set of known types. The resulting type names are 'type candidates'. (*iii*) The type candidates are aggregated across entities and automatically filtered based on their frequency and other checks, such as whether there is uncertainty in the posterior distribution over the type name. (*iv*) The remaining type names are manually filtered to exclude people types and file types. (*v*) The manually approved types are added to the type set and the process is repeated using this updated set. Iteration continues until the number of newly added types becomes sufficiently small to ensure that good overall coverage has been achieved.

This type discovery process works because entities usually have multiple types. For example, 'Tokyo' might be referred to as a project but also as a framework, system and toolkit. Each iteration of type discovery first finds entities whose types are any known type and then uses fact retrieval to add in the other types for these entities, which may be

unknown. Iteration of this process will discover any type, so long as there exists a chain of entities which connect that type to a seed type. In practice, given a large set of enterprise data, the process appears to give good coverage of types across a wide variety of domains. Specifically, on our largest dataset, it was able to discover a universe of 171 types across different domain that include "university", "case", "campaign", etc. Note that the EA's entity discovery process will surface only the subset of types that appear in the input dataset. For instance, types like "university" or "case" will be prevalent in the data belonging to an academic institution or a law firm, respectively.

## 4. Incremental Clustering

In our scenario, we assume that new documents will appear, or be updated, at different times and their overall volume will exceed the capacity of a single machine. Therefore, we designed EA to process documents *incrementally* and update the knowledge base in an online fashion. This architecture is illustrated in Figure 2.

In detail, let the set of documents available at time $t$ be $\boldsymbol{D}_t$. Given a set of templates $\boldsymbol{J}$, we use the Alexandria template matching system to produce a set of template matches $\{\boldsymbol{S}_{j,t}\}$ for $\boldsymbol{D}_t$. These template matches are split into $I$ batches $1, ..., I$ of arbitrary size so that $\boldsymbol{S}_{j,i,t}$ is the *i-th* batch available at time $t$ from the template set $\boldsymbol{J}$. For simplicity of notation, we focus the description on a single time interval, as it trivially generalizes to any time interval, and so we drop $t$ from the variables.

*Batch clustering*: Using the probabilistic program, the system takes $\{\boldsymbol{S}_{j,i}\}$ as observations and applies probabilistic inference to produce a set of discovered entities $\boldsymbol{E}_i$. For example, a document D1 titled "Project Tokyo overview" authored by Alice and a calendar event D2: "Tokyo team weekly sync" organized by Bob can be matched by structured templates containing the text templates "{Types} {Name} overview" and "{Name} {Types} weekly sync", respectively. Given these matches, batch clustering will output an entity {Name: Tokyo, Types: {Project, Team}, Members: {Alice, Bob}, Evidences: { D1, D2}}.

*Linking*: The entities discovered from the batch are linked to the knowledge base by the following steps:

i Query the knowledge base for candidate entities $\boldsymbol{Q}_i$, using a set of key properties, such as the entity name. To allow for name variants, the queries are extended to include different normalized forms of the entity names. For example, an entity with a name

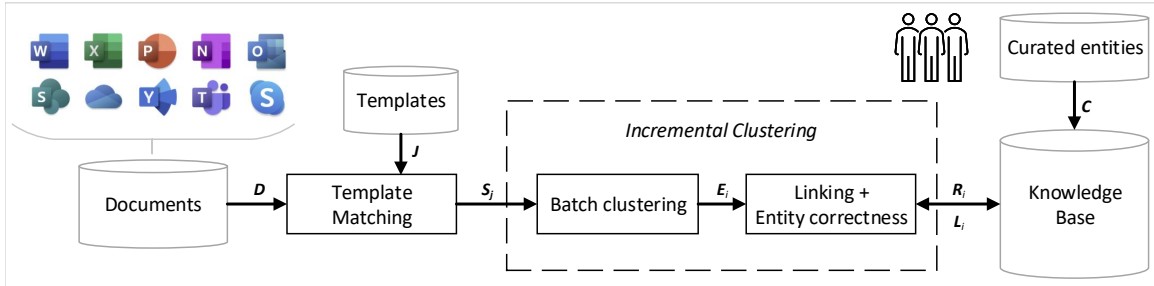

Figure 2: Architecture of the incremental clustering process used by EA.

"Cloud Storage Explorer" is queryable by "cloudstorageexplorer" and "CSE" keys which allows it to cluster incrementally with entities named "CloudStorage Explorer" and "CSExplorer" whose normalized names will overlap.

ii Produce a set of entities $\boldsymbol{R}_i$ by clustering together $\boldsymbol{E}_i$, $\boldsymbol{Q}_i$,

iii Update the knowledge base by replacing the entities $\boldsymbol{Q}_i$ with the result set $\boldsymbol{R}_i$ which contains both new and updated entities. For example, if the knowledge base contains an entity: "{Names: Tokyo, Types: {Project}, Members: {John}, Evidences: {D3}}", the linking step will conflate it with the entity produced from the batch clustering step and so create an updated entity "{Name: Tokyo, Types: {Project, Team}, Members: {Alice, Bob, John}, Evidences {D1, D2, D3}}".

*Curation*: During this process, the human curator can asynchronously edit the knowledge base and/or add new entities. Both the AI mined and curated entities are included in $\boldsymbol{Q}_i$ allowing mined and curated entities to be linked together into coherent merged entities. The batch size can be adjusted based on the memory constraints of the executor – a small batch size requires less memory but increases the runtime for processing all the batches.

*Entity Correctness*: In Alexandria, fact retrieval was performed for known entities which restricted the set of valid names for template matching. EA aims instead to discover entities and so template matching is unrestricted. This results in the system discovering entities which we do not want in the KB, such as people, months, locations, document names and tiny entities which are mentioned only in passing. To address this problem, we trained a 'correctness' model on a small sample of manually labeled entities extracted from eyes-on data. This model is a linear classifier, using simple features like the number of documents mined from, the kinds of documents, the template diversity and so on, that can be applied post-linking to estimate the correctness of each entity. Specifically, the classifier was trained on a sample of manually labelled 857 entities (274 positive and 583 negative labels) taken from the *Microsoft Eyes-off* dataset (see Section 6.1 for more details).

## 5. Related Work

The use of machine learning methods for large-scale automatic knowledge base construction has been intensively studied in the last decades [Dong et al., 2014, Carlson et al., 2010, Szekely et al., 2015, Zhang, 2015]. The majority of these existing methods are focused on Web-based information extraction, where the aim is to link new facts extracted from text to a set of entities coming from a public knowledge base such as Wikipedia [Dong et al., 2014] or Satori [Winn et al., 2019]; this task is often referred to as *entity linking* [Zhang, 2015]. Within an organization, entities are likely to be private rather than public, and so linking to public entities is not possible. Another line of related research on *entity discovery* deals with discovering new entities from scratch by processing text documents talking about these entities. Some of these methods perform *feature extraction* over the processed text to extract features characterizing mentions of a particular entity to guide the entity mining process. On such a basis, it is possible to conflate entity values found in mentions sharing similar feature patterns [Zhang, 2015]. In particular, the Named Entity Recognition (NER) features are widely used for this task as powerful signals for detecting categorised entity names in free text. Specifically, the NER models are able to detect entity names from the

text while also providing the type of each entity name chosen from a pre-defined set of types. To date, the most successful NER methods are based on neural language models, such as BERT (Bidirectional Encoder Representations from Transformers) and RoBERTa (Robust Optimized BERT Approach) [Devlin et al., 2019, Liu et al., 2019], that are trained on a large public corpus that can generalize to a private corpus with a small amount of fine tuning. While the categorized entity names extracted from NER models alone are not sufficient for constructing a full knowledge base, it is possible to adapt them to our task if an appropriate conflation algorithm is provided. Therefore, in the next section, we will consider the most competitive NER models as baselines for evaluating our approach.

## 6. Experimental Evaluation

This section evaluates the ability of EA to perform entity discovery, compared to four rival approaches:

**BERT Base Fine-tuned (BERT F)**  BERT is a state–of–the–art language model developed by Google that is commonly applied to NER tasks [Devlin et al., 2019]. It is pre-trained on a large corpus of English data in a self-supervised fashion and it makes use of 12 stacked layers of encoders with an attention mechanism to learn contextual relations between words in the text. This version of BERT is fine-tuned on the English version of the standard CoNLL 2003 dataset [Tjong Kim Sang and De Meulder, 2003] to recognize four types of entity names: location (LOC), organizations (ORG), person (PER) and Miscellaneous (MISC)[2].

**BERT Base Multi-lingual Fine-tuned (BERT MF)**  A variant of *BERT Base Fine-tuned* that is further fine-tuned on the CoNLL 2003 dataset to recognize the same four categories of entity names as BERT F in different languages beyond English[3].

**BERT Large Fine-tuned (BERT LF)**  This model similar to the above methods but it uses the large version of BERT with 24 layers of encoders and 16 attention heads. It is also fine-tuned on the English version of the CoNLL 2003 dataset to recognize 16 categories of entity names[4].

**RoBERTa Large Fine-tuned (RoBERTa LF)**  RoBERTa is a language model developed by Facebook that improves the performance of BERT using an optimized training strategy with a longer training epochs, bigger batches and a larger dataset [Liu et al., 2019]. This model is also fine-tuned on the English version of the CoNLL 2003 dataset[5].

The original Alexandria system [Winn et al., 2019] requires entity names as input and therefore not applicable to entity discovery.

All the benchmarks are freely available on the open-source huggingface platform. To make these models suitable for the entity discovery task, we must add an entity conflation model to conflate all the NER extractions referring to the same entity into a single entity.

---

2. huggingface.co/dslim/bert-base-NER

3. huggingface.co/wietsedv/bert-base-multilingual-cased-finetuned-udlassy-ner

4. huggingface.co/dbmdz/bert-large-cased-finetuned-conll03-english

5. huggingface.co/xlm-roberta-large-finetuned-conll03-english

Specifically, we use a standard case-sensitive linking by name conflation method that works as follows: Firstly, all extractions sharing the same name based on case-sensitive string matching are grouped together. Then, an entity is constructed by taking the union of all the documents' meta–data. For instance, $n$ extractions about an entity named $x$ classified with type $t_1, ..., t_n$ extracted from $j$ documents, $d_i, ..., d_j$, each with associated people's meta–data $m_i, ..., m_j$ (e.g. the authors of the documents) are conflated into the entity: "{Name: $x$, Types: $\{u(t_1, ..., t_n)\}$, Members: $\{u(m_i, ..., m_j)\}$, Evidences: $\{d_i, ..., d_j\}\}$" where $u(.)$ is the union function. To remove obvious noise, we filter out small entities from all the methods, i.e., with less than 100 evidences, and we also remove a few categories of entities that were not included in our ground truth, i.e., person, date and ordinal numbers. Finally, we use a list of disallowed entity names, containing 1998 stop-words and common English n-grams corresponding to false entity names, that is applied to filter the output of each method.

## 6.1 Datasets

Our evaluation involves the following three real-world datasets: *Enron*: A well-known public corpus of $517,401$ emails generated from 150 employees of the Enron Corporation [Klimt and Yang, 2004]. The dataset has full eyes-on access for research purposes (i.e., researchers can look at the data). *Microsoft Eyes-on*: A dataset of $368,366$ documents donated by Microsoft employees with restricted eyes-on access (i.e., the entities mentioned in the documents can be labeled by a selected pool of human judges). For this dataset, we also have access to a ground truth with 1,786 human-curated labels for real entity names (1,205 positive labels and 581 negative labels) that exist within the Microsoft organization. *Microsoft Eyes-off*: A larger dataset with $1,023,435$ documents made available by Microsoft employees with eyes-off access only that is useful for evaluating performance at larger scale. Both Microsoft datasets went through exhaustive privacy reviews in order to protect individual privacy and corporate confidentiality.

## 6.2 Performance Comparisons

We evaluate the performance of the alternative approaches against two versions of our method: ($i$) EA$_{size>100}$, that is our model with the basic entity size filter (i.e., $> 100$ evidences) that is applied to all methods. ($ii$) *EA*, that is our model with the entity correctness classifier described in Section 4. To evaluate the performance of the different approaches, we measure the precision and coverage against the ground truth defined as follows: Precision $= p/(p+n)$ and Relative Coverage $= p/p_{EA}$. Here $p$ and $n$ are the number of positive and negative entity name labels in the ground truth, respectively, that are calculated by matching in the entity names discovered by each method to the ground truth labels, and $p_{EA}$ is the precision of EA. Intuitively, the precision measures the proportion of discovered entity names with a positive label out of all the entities that can be matched to the ground truth, while the relative coverage measures the recall ratio between each method and EA. Absolute recall is not feasible to compute in our setting since it requires labels for entities that were not discovered by any system. Since *Enron* does not have any existing ground truth for entity names, for each method we drew a random sample of 100 entity names out of the names it discovered and manually labeled them positive or negative. This ensures unbiased precision estimates. For *Microsoft Eyes-on*, we used the existing ground truth.

|  | Enron | | | | Microsoft Eyes-on | | | |
|---|---|---|---|---|---|---|---|---|
| Method | Entities | Types | Prec. | Rel. Cov. | Entities | Types | Prec. | Rel. Cov. |
| BERT F | 2,764 | 3 | $0.08 \pm 0.04$ | $1.33 \pm 0.03$ | 990 | 3 | 0.44 | 0.33 |
| BERT MF | 5,789 | 3 | $0.13 \pm 0.03$ | $7.86 \pm 0.04$ | 2,610 | 3 | 0.39 | 0.77 |
| BERT LF | 2,296 | 16 | $0.18 \pm 0.04$ | $4.31 \pm 0.04$ | 867 | 16 | 0.58 | 0.38 |
| RoBERTa LF | 5,988 | 3 | $0.13 \pm 0.03$ | $8.13 \pm 0.04$ | 2,542 | 3 | 0.49 | 0.94 |
| $EA_{size>100}$ | 449 | **152** | $0.37 \pm 0.05$ | 1.73 | 1,029 | **141** | 0.71 | 0.55 |
| EA | 104 | **152** | $\mathbf{0.92} \pm 0.03$ | 1 | 1,591 | **141** | **0.83** | 1 |

Table 2: Performance of all the methods on the Enron and Microsoft Eyes-On datasets.

In detail, Table 2 reports the performance of the five methods on the *Enron* and *Microsoft Eyes-on* datasets. In these experiments, the EA methods used a set of 88 templates and hierarchy of 176 entity types, both learned from the *Microsoft Eyes-off* dataset. On both datasets, $EA_{size>100}$ and EA have the highest precision. In particular, the precision of EA is 5.11 (0.92 vs. 0.16 on *Enron*) and 1.43 (0.83 vs. 0.58 *Microsoft Eyes-on*) times higher than the second best method (BERT LF). This is due to the combination of the template-based language model and the probabilistic clustering of EA that allows it to jointly discover entities with high precision and disambiguate different entities with the same name. For example, EA was able to find two distinct entities named 'MT6' on *Microsoft Eyes-on*, while the same entities are incorrectly over-conflated together due to linking entities by name in the other methods. In addition, through its name variant model (see Section 3.2), EA was able to find alternate names for several entities, such as {OneDriveWeb, OneDrive Web, OD Web} in *Microsoft Eyes-on* and {Phillips, PHILLIPS} in *Enron*, referring to the different ways that people expressed the same entity in the free text. Instead, the same entity names were incorrectly under-conflated by the other methods because they do not strictly match with one another. In terms of coverage, EA has the highest coverage on *Microsoft Eyes-on* while the BERT-based methods have higher coverage but lower precision on *Enron*. It is worth noting that in our application, given the presence of the knowledge curator that must editorially check the validity of each entity, we find that methods with high coverage and low precision are less practical because they tend to overload the system with a very large and noisy set of entities, more than a human curator can cope with. Instead, high-precision methods like EA tend to provide a more manageable, yet substantial, set of entities that are much easier to deal with for the knowledge manager. In this light, precision errors tend to be penalized more than providing additional coverage. Finally, EA is able to associate up to 9.5 times more types (152 vs. 16 on *Enron*) to the discovered entities compared to the other methods that can only associate types from their (typically small) set of types that they have been fine-tuned on. Overall, EA achieves significantly higher precision and it retrieves more entity types on both datasets.

### 6.3 Memory Usage and Runtime

To assess the performance of EA at larger scale, we ran it on the *Microsoft Eyes-off* dataset. For compliance reasons, we could not run the other methods on this dataset. In this experiment, EA was able to process 1,023,435 documents and 45,645,618 template matches in 9.3h on a conventional machine, using 8.64GB of memory measured as the median of the last 50 batches (Memory@50). It discovered 675,439 entities and 172 entity types. The

clustering time is constant across batches, taking approximately 23 seconds for a batch of 10,000 template matches. The linking time increases approximatively linearly with a small slope (0.005) over batches as the size of the knowledge base grows over time (that can potentially be mitigated by cleaning up old entities). Overall, this result shows that the incremental clustering architecture adopted in EA scales gracefully on our largest dataset.

### 6.4 Type Retrieval

Figure 3 shows the types retrieved by EA on the two eyes-on datasets. Due to the lack of ground truth for types, we cannot perform a quantitative evaluation of their correctness. However, since it is known that *Enron* and *Microsoft* are companies specialized in the energy and software industry, respectively, the types appear relevant to the organization owning the data (e.g. "Site" or "Fund" for *Enron* and "Team", "App" or "Deal" for *Microsoft Eyes-on*). When we focus on the 'Technology' and 'Seed types' classes from the user study (Table 1), we found that 40% of the entity types declared by the participants were also retrieved by EA for at least one entity in the *Microsoft Eyes-on* dataset. The non-retrieved types are mostly specific categories such as "Best practice", "O365" or "Service measurement" that did not appear often in the training set. As a result, we expect a substantial but not necessarily complete overlap between the two sets, even if they both belong to the technology industry.

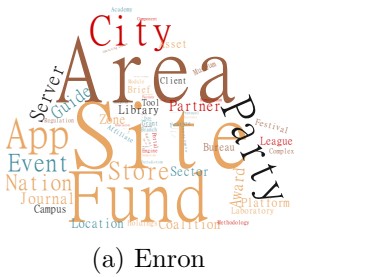

(a) Enron

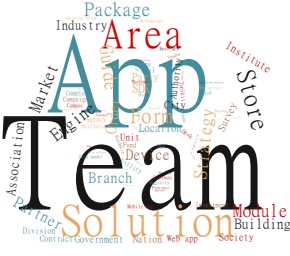

(b) Microsoft Eyes-On

Figure 3: Word clouds of entity types retrieved by EA on the *Enron* and *Microsoft Eyes-on* datasets. The font size is weighted by frequency of each type across all entities.

### 7. Conclusions

Enterprise Alexandria provides automatic knowledge discovery of typed entities from enterprise documents. The system extends Alexandria by consuming structured information coming from documents' metadata, discovering entities incrementally in an online fashion, and allowing users to contribute manual curations to the entity discovery process. Our evaluation shows that EA significantly outperforms the alternatives in extracting entities with higher quality on several real-world datasets. EA is a key technology for the AI engine of Microsoft Viva Topics and it is currently servicing thousands of organizations and users[6].

---

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
