# OpenReview forum: "Enterprise Alexandria: Online High-Precision Enterprise Knowledge Base Construction with Typed Entities"
_AKBC.ws/2021/Conference — AKBC 2021_

### Official Review · Reviewer_fWqx · 2021-07-02
**lack of focus, unclear contributions and difficult to read and follow**

**Rating:** 4
**Confidence:** 5

**Review:**

The paper describes a system named “Enterprise Alexandria” for knowledge base construction with typed entities. The approach uses relevant methods.
Overall, the paper is difficult to read and follow. It is very difficult to decode the main contributions of the paper. Some parts are very briefly described and it is difficult to connect and understand different parts of the paper. The evaluation is also questionable - 1) the focus is unclear - what has been evaluated?, and 2) the comparison/experiments with the related methods improperly designed.
In summary, there are some contributions but they are not well presented. The paper is a mix of presentation of a system/approach/workflow and description of few methods -> the paper lacks focus. E.g. it would be better if the paper describes the system only, or focus on a particular method, e.g. entity linking or entity typing. In the current form it is a mixture and confuses the reader.
Considering my comments above I do not recommend acceptance of the paper.
 More comments below.

==

= Abstract

“ability to process both the textual information and the structured metadata available in each document in an online fashion“
-> it is not clear what is meant by “online fashion”

“The knowledge discovery process uses a probabilistic program defining the process of generating the data item from a set of unknown typed entities.”
-> complex sentence

Section 4. Incremental clustering
-> it is unclear the purpose of the entity correctness task. It is applied, but it is unclear if any follow-up action is then. Also, there are missing details about how it actually works.

Section 6. Experimental Evaluation
-> It is unclear what exactly has been evaluated? I.e. what part of the pipeline? Typing, linking, matching?
-> the setup of the experiments are questionable. The (BERT) models have been trained and perform typing with just a few types (3, 16 types), while the model proposed in the paper performs typing of over 152. The comparison is unbalanced. More proper setup would be to train/configure the BERT to perform typing with the same amount of types - that would be fair comparison.

Section 6.4. Type Retrieval
-> Instead of word cloud figures the authors could consider providing in-depth analysis on the performance of entity typing.

---

> ### Author Response · Authors · 2021-07-29
> **Response to reviewer 4**
>
> Thanks for the review.
>
> To clarify some potential misunderstandings and answer some questions raised, “online” stands for online machine learning and it is a standard term used in ML (https://en.wikipedia.org/wiki/Online_machine_learning) referring to “a method of machine learning in which data become available in a sequential order”, just like in EA. On the entity correctness model, as stated in the last paragraph of Section 4, the entity correctness model is used to filter out entities that are not required for the knowledge base, e.g., dates, locations, document names and tiny entities.
>
> Regarding section 6, fine-tuning BERT on 152 types would require ground truth labelling of 152 types.  No such ground truth is available, and EA doesn’t need it.  In fact, EA discovered the 152 types without any ground truth.

---

### Official Review · Reviewer_2sAy · 2021-07-06
**Well-motivated real-world problem, the main contributions are nicely explained.**

**Rating:** 8
**Confidence:** 3

**Review:**

The authors have presented a dedicated new system for automatically constructing a knowledge base from private enterprise data. This system is an extension of an existing system designed for Web-based information extraction.  It employs a probabilistic program to generate both the document and its metadata from a set of latent entities and adds the ability to learn the set of types associated with each entity from a pre-defined type hierarchy.  . This work also presents a real-world evaluation showing that the developed system scales well and significantly improves the precision of the extracted knowledge compared to four baselines on three real-world datasets.


Strengths:

 - the problem this work handles is well-motivated, and the paper is well written and easy to follow.
- the authors explain very clearly their contributions, challenges, and novel extensions to Alexandria.
- the presented system supports online settings, where the documents are uploaded not at a single time point and the memory consumption exceeds the capacity of a single machine, which is a real-life scenario for enterprise data.
- the authors compare their proposed system with state-of-the-art baselines, showing their advantages in real-life scenarios.

Weaknesses:

- the presented user study is very interesting and nicely demonstrates
why a wide variety of entity types are important. However, this study includes a very small number of participants, and important details regarding the grouping of the categories are missing.
-It would be of interest to evaluate each component's marginal contribution, e.g., using other clustering methods to better explain the reviewers' choices.

---

> ### Author Response · Authors · 2021-07-29
> **Response to Reviewer 3**
>
> Many thanks for your comments.
>
> Please see our replies to reviewer 1 and reviewer 2 above about the importance of the user study. The small number of participants is consistent with qualitative research, which is designed to look for patterns of meaning rather than quantities. In terms of method, the categories that were defined by participants were grouped inductively (bottom-up) using an affinity diagram, and groupings were iterated on based on feedback from collaborators. The groupings themselves were based on the category names, descriptions and discussions provided by participants in the workshop. The categories were grouped primarily at the semantic level, which is in the explicit and surface meanings of the data; for example, participants used terms such as “various tools during projects”, “insightful information”, and “people working in org” in their descriptions and discussions. However, as outlined in our response to Reviewer 2, we also examined latent (underlying) patterns of meaning to demonstrate similarities and differences across the groupings that were generated through the affinity diagram exercise . We have added more details to the user study subsection about how categories were grouped, which we will expand in the extended version of the paper.
>
> Finally, due to space limitations, we were not able to add individual evaluations for each component to complement the end-to-end evaluation of EA reported in the paper. We will aim to add these evaluations in an extended version of the paper.

---

### Official Review · Reviewer_Pg7f · 2021-07-18
**Interesting and useful System that needs more work and clarifications on the experimental side**

**Rating:** 6
**Confidence:** 4

**Review:**

This work extends the Alexandria system for KB construction [Winn et al.,19] , to the particular scenario of constructing enterprise-specific knowledge bases. The main system extensions described in the paper are:
(1) Extending Alexandria’s templates mechanism to support (semi-)structured documents, that are common in enterprise scenarios such as emails & calendar events. This allows the system to leverage the availability of semi structured data when building the KB.
 (2) Improved entity disambiguation mechanism which considers the number and length of words, as well as variants stemming  from reasonable enterprise scenarios
(3) Type discovery mechanism - The original entity discovery process described in [Winn et al.,19] is extend to  allow the system to partially discover types. This is very practical in the Enterprise scenario, where the entity types may vary from enterprise to enterprise, as oppose to the case of general KBs.  However, the new types are learned for entities with already known types, so this does not extend to the general case of arbitrary type discovery.
Also, An additional incremental computation mechanism is introduced which allow to incorporate manual annotation/curation


Overall this work showcase and interesting and useful extension of Alexandria, although the technical contributions seem incremental and without much technical depth. The main issue with this work is that the experimental evaluation section requires more clarifications and experiments.

Strong points:

S1 The paper extends Alexandria to particularly construct an enterprise-specific knowledge base, which is an interesting, practical use case

S2 The extensions proposed in Section 3 takes the main differences between the general KB and the enterprise KB.

S3 The Experiments section provides an interesting comparison between the dedicated approach proposed in the paper with with several baselines for general-purpose NER.

Weak Points (refer to the Detailed Comments Section for more information)

W1 The introduction and related work section are somewhat confusing and do not properly justify the particular work and the contribution made.

W2 The qualitative study does not seem informative and useful in the rest of the paper

W3 The extensions proposed to Alexandria do not demonstrate much novelty and technical depth

W4 The experimental section lacks several important clarifications and additional experiments, as detailed in commends D4 and D5


Detailed Comments:

D1 Positioning: The contribution is not clearly stated in the introduction, as the authors first claims that EA is specifically designed to answer a privacy constraint that forbids manual annotation of enterprise data. However in the contribution section the authors do not address the privacy issue anymore. Also, the abstract should state that EA extends Alexandria.

D2 The purpose of the qualitative study is not clear. The authors asked the participants to list typed entities relevant to their organization, and conclude that “a wide variety of entity types are important for work”. What exactly is wide? Are all these entities necessary? How is this a conclusion from the study? And in any case,  48 entity types obtained for two different organizations, sounds really small in comparison with general knowledge bases (e.g. dbpedia) which contains thousands of entities.

D3 While the extensions of Alexandria  seem appropriate to the enterprise scenario, they do not show great technical depth and novelty.

D4 The authors compared their approach to several baselines based on pre-trained language models that were fine tuned for NER tasks with limited, predefined number of entity types. If we are interested in specific types, isn’t it better to perform the fine tuning over some more relevant types? This may greatly improve the precision of these baselines.

D5 Several Aspects in the experiments are not clear:
(i) Which particular entities were labeled in the datasets? Of which/ how many types? What are the negative entity examples? As one dataset is anonymous and the other is Enron + manual labels, this requires clarification
(ii) What is the relative coverage measure and why was it used instead of standard measures such as  recall?
(iii) Why does the EA version with >100 evidence obtained lower precision scores? Intuitively it should be more “correct” w.r.t. the entities it identifies, on the account of the recall.
(iv) Why wasn’t Alexandria a baseline in your experiments? It seems reasonable to show that the existing system does not properly identify the desired entities and types in an enterprise data scenario.
(v) The evaluation of the the type retrieval mechanism, one of the major contributions of the paper, is lacking, to my opinion, as it only shows a world cloud to exemplify the discovered types. There should be an experiment that measure the quality/utility/correctness of the types discovered.
(vi) The details regarding the performance of the system, which rely on the incremental/clusterting mechanism of Section 4, are lacking a comparison with the approach described in [Winn et al.,19]  in order to demonstrate that the optimization proposed in the paper is indeed beneficial.

---

> ### Author Response · Authors · 2021-07-29
> **Response to Reviewer 2**
>
> Many thanks for the insightful review. On the comment regarding the positioning of the paper, as you correctly state, the introduction does mention 1) the privacy constraints and 2) the lack of manual annotations in building a knowledge base for an enterprise as two key differentiating factors. These requirements are respected throughout the design and the application of EA. In fact, the model training is unsupervised except for a small number of seed types injected in the type discovery process and the small sample of manually labelled entities (857 labels, 274 positives, 583 negatives) used to train the entity correctness model. Respecting these two requirements makes EA the first method that is truly applicable and competitive in Enterprise scenarios to the best of our knowledge, which makes it a relevant contribution to the state of the art. We have clarified this part in the introduction as well as added more details to the size of the training set of the entity correctness model. We also agreed about stating that EA is an extension of Alexandria in the abstract and we have modified the revised version accordingly.
>
> Regarding the contribution of the user study, please see the first paragraph of the reply to Reviewer 1. Regarding the "wide variety" of entity types, the variety was in terms of relative meaning and purpose for the user, rather than quantity in comparison to general knowledge bases (see continued response in a separate comment below).
>
> Regarding improving the NER baselines with fine-tuning on specific types, in principle we agree this can improve their precision but it is challenging to find labels for types to train on in an enterprise setting - in fact we have none. This feature further highlights the ability of EA on learning types without requiring such labels.
>
> To clarify the concerns raised in D5:
> -	(i) As reported in Section 6.2, “we manually labelled a random sample of 100 entity names for each method”.  The labels were either positive or negative.  Entity types were not manually labelled and their correctness was not evaluated. We have now clarified this point in the revised version (see Section 6.4).
> -	(v) To further expand on the evaluation of types retrieval mechanism, an empirical evaluation of the correctness of the types requires additional ground truth, ideally to be sourced from employees of each organization. Unfortunately, we do not have access to any ground truth for types. However, since it is known that *Enron* and *Anonymous* are specialized in the energy and software industry, respectively, -- the latter will be apparent in the final version when the actual name for “Anonymous” will be disclosed -- it is possible to see that the top discovered types in each dataset, including “Site”, “Area” and “City” for *Enron* and “Team”, “App” and “Package” for *Anonymous* are very relevant to their business. We have now discussed the lack of ground truth in the experimental section.
> -	(ii) As reported in Section 6.2, “the relative coverage measures the recall ratio between each method and EA”.  It was used instead of recall because recall requires collecting additional ground truth for all entities, including those that were not discovered by any system.  We have now clarified this point in the revised version.
> -(iii) As explained in Section 6.2, EA(size>100) is a version of EA that lacks the entity correctness classifier. Regarding the difference in precision between (size>100) and EA, the size filter can be regarded as a basic entity correctness model using only one feature (i.e., size). Therefore, it is expected to be outperformed by the more sophisticated entity correctness model used by EA that has access to more features “like the number of documents mined from, the kinds of documents and the template diversity” (see Section 4)
> -(iv) Alexandria cannot be a baseline for our task because, as presented in its original paper (Section 4), Alexandria deals with three different tasks: template learning (i.e., discover a set of templates from text given a set of known entities), schema learning (i.e., discover new properties from text given a set of entities and templates) and fact retrieval (i.e., new facts about entities are retrieved from text given the entity names). Also, Alexandria had access to a set of 3000 real entity names that we don’t have for our dataset. Making Alexandria suitable to jointly solve the entity discovery and type discovery tasks requires a set of non-trivial extensions that we described in the paper. In summary, EA has a capability of entity and type discovery that Alexandria didn't have.

---

> > ### Author Response · Authors · 2021-07-29
> > **More Details on User Study**
> >
> > To further expand on the “wide variety” of entity types, the variety was in terms of relative meaning and purpose rather than quantity in comparison to general knowledge bases. To demonstrate the variety of meaning, the categories described by participants varied in terms of:
> >
> > - **Physical referent**, from abstract (e.g., management buy-in, industry knowledge) to concrete (e.g., papers/reports, tasks);
> > - Similarly the **audience** for the categories varied from knowledge intended for ‘insider’ team members (e.g., tasks, budget), enterprise wide (e.g., design library, documentation), and ‘outsiders’ who are external to the enterprise (e.g., media, research papers).
> > - **Focus**: What something is (e.g., stakeholder, research template) versus what something does (e.g., communicate)
> > -  **Temporality**: Whether the category was largely fixed (e.g., headcount) or fluid - in terms of potential for additional items (e.g., research papers, tools) or changes to the item itself (e.g., problems, roadmap) – over time.
> >
> > In turn, it becomes relevant to ask:
> >
> > 1. Whether and how this variety of categories can be meaningfully extracted from enterprise data and represented as types?
> > 2. What are the opportunities for users to partner with Enterprise Alexandria to capture these different layers of meaning?
> >
> > Due to limitations of space, we will provide details about this deeper analysis in the extended version of the paper.

---

### Official Review · Reviewer_xrNy · 2021-07-19
**Challenging, real world problem with evaluations that could be better**

**Rating:** 6
**Confidence:** 3

**Review:**

Summary: Enterprise Alexandria automatically builds knowledge bases with typed entities from private enterprise datasets. To build a KBC and identify typed entities, EA uses a probabilistic program and does not use meta-data and human annotations, which are generally not available in enterprise datasets anyways.

The authors ran a user study and they show that users need a variety of entity types and users wish to engage with different types in different ways. The authors claim that the study helps in the design of EA's user model.

Evaluations in EA focused on two aspects: (1) the precision and coverage of EA's identified entity types  against the precision and coverage of outputs from 4 BERT-based models and (2) the scalability of EA's incremental clustering. The results show EA has gains in terms of precision.


Strong Points:
1. Explains the challenges, motivation, and novelty fairly well.
2. Evaluations proved gains in precision with EA compared to BERT models, which are the state-of-the-art methods.
3. Performance evaluation proved how well the incremental clustering scaled in terms of memory usage and runtime.
4. The world cloud visualization shows how relevant the most frequent entity types in the dataset are to the dataset topics. Also, 40% of the user-expected types (from the user study) were retrieved by EA.


Weak points:
1. It's not clear how the user study contributes to the design of EA itself.
2. Don't entity types differ from one dataset domain to another? The results show that 40% of the user-expected types were retrieved by EA, which is great, but the domains covered from the user study (entity types related to multinational corporation data and types related to data found in the tourism and travel industry) are different from the domain datasets from the evaluation (types expected of Enron and Anonymous Eyes-on). How were the differences in domain addressed when comparing the types from EA and the types identified by the users from the user study?
3. While there are gains for precision across both datasets in the evaluations, it's not the same for relative coverage. It would be interesting to see how relative coverage fares on a third dataset (if it is even feasible to find such a third dataset).
4. Word cloud visualizations and the datasets associated need a bit more context. A few sentences describing what datatypes are expected based on a given dataset would help.

Overall, the paper coupled with the evaluation results are promising. However, there are certain aspects of the paper that needs to be addressed.

---

> ### Author Response · Authors · 2021-07-29
> **Response to Reviewer 1**
>
> Thanks for the detailed review and the important points raised. Regarding the contribution of the user study, as stated both in the introduction and in the incipit of Section 2, this study contributes to support the design and the evaluation of Enterprise Alexandria. Its findings are essential to justify the need for discovering types as part of the entity discovery task in the enterprise setting. Since type discovery is a non-trivial task that adds extra complexity to our model, it is important to justify why it is essential through a qualitative user study, which indeed supports this requirement. It also provides reliable data about real entity types for two specific organizations that are important for our evaluation.
>
> We do expect that entity types will vary across different domains and Enterprise Alexandria is able to accommodate this requirement. To clarify, as stated in the last paragraph of Section 3.3, the iterative type discover process leads to discover all types that have a chain of entities that connects that type to a seed type. On our largest dataset, this process led to discover a large universe of types (precisely 171 types including “university”, “case”, “campaign”, etc.) that appear to represent a wide variety of domains. Then, the entity discovery process will surface only the subset of types that appear in the input dataset. In practice, this means that types like “university” or “case” will be prevalent in the data belonging to an academic institution or a law firm, respectively.  We have now clarified this point in the revised version.
>
> The above also explains the 40% overlap between the types discovered in *Enron* and *Anonymous Eyes-on* and the types provided by the user study. Since both *Enron* and *Anonymous* are technology companies (the latter will be apparent when the name will be revealed in the final version), we cannot expect full overlap with the types reported by the participants given they come from different domains. However, as stated in Section 6.4 when we focus the “Technology” and “Seed” types, there is a significant overlap with the types discovered by EA, while the non-retrieved types tend to be more specific categories that we do not necessarily expect to be retrieved in every dataset belonging to the technology industry. We have clarified this point and improved the description of the world cloud in the revised version.
>
> Finally, on the higher relative coverage but lower precision of the baselines w.r.t. EA, unfortunately we do not have access to a third dataset to further validate this result. Intuitively the results suggest that the BERT-based models discover a very large set of entities, but their precision is significantly lower compared to EA. In our application, however, given the role of the knowledge curator, who must editorially check the validity of each entity (see first paragraph of Section 1), we find that methods with high coverage and low precision are less practical because they tend to overload the curator with a very large and noisy set of entities, more than a human editor can cope with. Instead, high-precision methods like EA tend to provide a more manageable, yet substantial (i.e., >100 entities for Enron >1000 entities for Anonymous Eyes-On), set of entities that are much easier to deal with for the knowledge manager. In this light, errors in precision should be penalized more than providing additional coverage. We have clarified this point in the evaluation section.

---

### Decision · Program_Chairs · 2021-08-17

**Decision:**

Accept

**Comment:**

The reviews are mixed and some of the concerns, mainly ones that are related to the presentation, remain after the authors' response. Due to the paper's strengths (good motivation, interesting system, promising even if imperfect evaluation), we have decided to recommend acceptance. Please carefully follow the reviewers' comments to improve the writing for the camera ready version.